

# Pharmacokinetics and safety of oral glyburide in dogs with acute spinal cord injury

Nick Jeffery[1], C. Elizabeth Boudreau[1], Megan Konarik[2], Travis Mays[2] and Virginia Fajt[3]

[1] Department of Small Animal Clinical Sciences, Texas A&M University, College Station, TX, United States of America
[2] Veterinary Medical Diagnostic Laboratory, Texas A&M University, College Station, TX, United States of America
[3] Department of Veterinary Physiology & Pharmacology, Texas A&M University, College Station, TX, United States of America

## ABSTRACT

**Background**. Glyburide (also known as glibenclamide) is effective in reducing the severity of tissue destruction and improving functional outcome after experimental spinal cord injury in rodents and so has promise as a therapy in humans. There are many important differences between spinal cord injury in experimental animals and in human clinical cases, making it difficult to introduce new therapies into clinical practice. Spinal cord injury is also common in pet dogs and requires new effective therapies, meaning that they can act as a translational model for the human condition while also deriving direct benefits from such research. In this study we investigated the pharmacokinetics and safety of glyburide in dogs with clinical spinal cord injury.

**Methods**. We recruited dogs that had incurred an acute thoracolumbar spinal cord injury within the previous 72 h. These had become acutely non-ambulatory on the pelvic limbs and were admitted to our veterinary hospitals to undergo anesthesia, cross sectional diagnostic imaging, and surgical decompression. Oral glyburide was given to each dog at a dose of 75 mcg/kg. In five dogs, we measured blood glucose concentrations for 10 h after a single oral dose. In six dogs, we measured serum glyburide and glucose concentrations for 24 h and estimated pharmacokinetic parameters to estimate a suitable dose for use in a subsequent clinical trial in similarly affected dogs.

**Results**. No detrimental effects of glyburide administration were detected in any participating dog. Peak serum concentrations of glyburide were attained at a mean of 13 h after dosing, and mean apparent elimination half-life was approximately 7 h. Observed mean maximum plasma concentration was 31 ng/mL. At the glyburide dose administered there was no observable association between glyburide and glucose concentrations in blood.

**Discussion**. Our data suggest that glyburide can be safely administered to dogs that are undergoing anesthesia, imaging and surgery for treatment of their acute spinal cord injury and can attain clinically-relevant serum concentrations without developing hazardous hypoglycemia. Serum glyburide concentrations achieved in this study suggest that a loading dose of 150 mcg/kg followed by repeat doses of 75 mcg/kg at 8-hourly intervals would lead to serum glyburide concentrations of 25–50 ng/mL within an acceptably short enough period after oral administration to be appropriate for a clinical trial in canine spinal cord injury.

Corresponding author
Nick Jeffery, njeffery@cvm.tamu.edu

## INTRODUCTION

Despite extensive research on the mechanisms mediating the transition from traumatic spinal cord injury to tissue destruction, an unequivocally-accepted medical intervention for reduction in tissue destruction has not been introduced into clinical medicine. This failure is especially surprising bearing in mind the large number of interventions that have been effective in experimental rodent models (*Kwon et al., 2011*; *Tetzlaff et al., 2011*). For that reason, it appears that, rather than a lack of discovery of putative interventions, the roadblock to introduction of new therapies lies in translating new therapies from laboratory to clinic.

There is a 'large translational gap' between rodent experiments and human clinical practice meaning that testing in additional models is required to bolster confidence in taking forward specific interventions to human clinical trials (*Kwon et al., 2015*). There is a role for many models to achieve this aim: for instance, laboratory models of spinal cord injury in primates are important in investigation of strategies to improve hand use (*Salegio et al., 2016*) and pig models are useful for making measurements of intradural pressure following experimental spinal cord injury (*Streijger et al., 2017*). However, an important difference between laboratory models and human spinal cord injury is the heterogeneity of clinical patients; notably they vary in genetics, age, comorbidities and time delay between injury and access to therapy. Such heterogeneity introduces several sources of variation into outcome measures and implies that apparent effectiveness of an intervention in the laboratory may not be apparent when applied in clinical patients. It is difficult to model this disparity using laboratory animals but spinal-injured veterinary patients mimic both lesion and treatment of human spinal cord injury. Pet dogs show considerable heterogeneity (similar to human patients), commonly incur spinal cord injury in the course of their everyday lives and undergo similar diagnostic and therapeutic interventions as human patients, with similar limitations in their recovery rates (*Moore et al., 2017*). Therefore, there are many benefits to investigating putative therapies for spinal cord injury in canine veterinary patients. In addition, from a veterinary perspective, it is also necessary to identify new therapies for the dogs that incur such injuries.

Glyburide (also known as glibenclamide) has been widely studied in treatment of many CNS lesions in the laboratory, including stroke (*Simard et al., 2006*), traumatic brain injury (*Simard et al., 2009*) and spinal cord injury (*Simard et al., 2007*; *Simard et al., 2012*; *Popovich et al., 2012*). Glyburide's mechanism of action in CNS injury is via blockade of the Sur1-Trpm4 channel that is highly upregulated after CNS injury, particularly in blood vessels (*Simard et al., 2007*; *Gerzanich et al., 2009*). The Sur1-Trpm4 channel allows ingress of cations into affected cells, resulting in cytotoxic edema and cell lysis (*Simard, Kahle & Gerzanich, 2010*). Glyburide binds to Sur1 at very low concentrations (10–100 nM; *Aguilar-Bryan et al., 1990*), especially at the low pH found in ischemic tissue (*Simard et al., 2008*) and its effects compare well with other drugs that might aid in preserving damaged

spinal cord after insult when tested in rodent models (*Hosier et al., 2015*). Furthermore, it is an attractive candidate for translation because its pharmacological effects are well-known, a dose range to achieve therapeutic levels has been calculated for human clinical trials (*Sheth et al., 2016a*), the drug is in widespread use (in humans) for treatment of type II diabetes (*Rendell, 2004*) and it is widely available as an inexpensive oral preparation. Pancreatic beta cells also express Sur1 receptors (*Panten, Schwanstecher & Schwanstecher, 1996*) meaning that the drug effects on spinal cord endothelial (and other) cells are mediated via the same mechanism as that on blood glucose. Therefore, the main safety concern with use of this drug for spinal cord injury is hypoglycemia.

The safety of oral glyburide in healthy dogs has been evaluated previously (*Guan et al., 2014*; *Liu et al., 2014*; http://products.sanofi.ca/en/diabeta.pdf). In this study we wished to extend these assessments into a cohort of dogs that had spinal cord injuries for which they were undergoing investigation and surgery because these other interventions and the lesion itself may alter blood drug concentrations or risk of adverse effects. For instance, drug absorption can be compromised by the reduction in gut motility associated with anesthesia (*Torjman et al., 2005*) or spinal cord injury itself (*Cruz-Antonio et al., 2006*; *Cruz-Antonio et al., 2012*). Specifically, our objectives were to evaluate the effects on blood glucose of glyburide and to assess drug disposition, in order to make recommendations for dose regimens for a subsequent clinical trial in dogs with similar injuries.

## MATERIALS & METHODS

We recruited pet dogs with acute, naturally-occurring spinal cord injury. Dogs affected in this way routinely undergo general anesthesia, cross-sectional imaging, decompressive spinal surgery, and they receive intravenous fluid infusions, any of which might affect drug disposition or the effects of glyburide on blood glucose. Animals were treated in this study as they would be clinically, for instance, intravenous fluid was given according to each individual's requirements as determined by the attending anesthetist, because we were interested in mimicking future clinical application of glyburide. There were several phases to this study: (i) testing whether oral glyburide caused hypoglycemia in the target patient population; (ii) determining concentrations of glyburide over the 24-hour period following a single oral dose of glyburide administered just before induction of anesthesia for imaging and surgery; and, (iii) estimating a dose regimen that would achieve a plasma concentration of glyburide between 25 and 50 ng/mL for at least 72 h (*Simard et al., 2008*; *Sheth et al., 2016a*; *Sheth et al., 2016b*).

### Animals

Dogs included in this study presented for treatment of acute spinal cord injury to the small animal hospital of the Colleges of Veterinary Medicine at Iowa State University or Texas A&M University, and study protocols were approved by the relevant Institutional Animal Care and Use Committees (Iowa State, log number: 1-16-8148-K; Texas A&M, IACUC 2016-0324 CA, reference number 044949). For inclusion, dogs had to have become acutely non-ambulatory with suspected intervertebral disc herniation occurring less than 72 h previously and be ready to undergo general anesthesia and surgery (i.e., fasted); those that

were thought likely to have lesions other than intervertebral disc herniation, or that were not likely to undergo imaging and surgery, were not invited to take part. Dogs in either of these categories that were initially included (i.e., given the drug) were subsequently excluded from further analysis. Dogs diagnosed with hyperadrenocorticism or diabetes mellitus were excluded.

## Materials

For glyburide dosing we obtained commercially available generic glyburide tablets of 1.5 mg (Teva) and 1.25 mg (Heritage) each. Blood glucose was measured using a commercially-available glucometer and test strips designed for diabetic monitoring in dogs (AlphaTrak2; Abbott Laboratories, Abbott Park, IL, USA); the performance of this testing method against reference standards has been previously published (*Cohen et al., 2009*).

For measurement of glyburide concentration in blood samples, control normal canine plasma was obtained from Equitech-Bio, Inc. (#SCAPE35-0100; Kerrville, TX, USA). Glyburide (USP Reference, #1295505) and Glipizide (G117) reference materials were obtained from Sigma (St. Louis, MO, USA). PBS Buffer, pH 7.0, was made using 8 g NaCl, 0.2 g KCl, 1.44 g $Na_2HPO_4$, and 0.24 g $KH_2PO_4$ dissolved in 1 L reverse-osmosis deionized (RO-DI) water, pH adjusted to 7.0 with a 6N HCl solution, sourced in-house. All chemicals and reagents were ACS grade and obtained from VWR Scientific (Randor, PA, USA).

## Methods

### Safety study

Five dogs that were paraplegic and underwent imaging and surgery under general anesthesia were included; in this part of the study there were no dog weight restrictions applied. After receiving informed owner consent, each dog received 75 mcg/kg glyburide orally; the dose was chosen based on previous safety studies in dogs (http://products. sanofi.ca/en/diabeta.pdf) and the dosages given in similar human clinical trials (https://www.clinicaltrials.gov/ct2/show/NCT02524379?term=glyburide+spinal+cord&rank=1; *Sheth et al., 2014*). Partial tablets were used as appropriate for the weight of the dog. At hourly intervals from 0 to 4 h, and 2-hourly intervals from 4 to 10 h after drug administration, a single drop of blood was withdrawn by needle puncture of a peripheral vein and tested for blood glucose concentration using a blood glucose test strip (AlphaTrak2). If a value of less than 50 mg/dL was recorded then that individual received intravenous glucose as required to normalize the glucose to within the laboratory reference interval (76–119 mg/dL).

Anesthesia, surgery and ICU technicians were asked to record any observed adverse effects that occurred at any time (with direction to pay particular attention to signs of ataxia, disorientation, or other potential signs of hypoglycemia), and notes were made of any need for supplementary glucose administration. (It should be noted that blood glucose is not routinely monitored in dogs undergoing anesthesia and surgery.)

### Pharmacokinetic study

Six dogs weighing more than 5 kg and undergoing imaging and surgery for treatment of acute intervertebral disc herniation resulting in non-ambulatory paraparesis or paraplegia

were included in this part of the study. As before, after obtaining informed consent, each dog was given 75 mcg/kg glyburide orally as a tablet or broken tablet. Blood samples (1 mL) were obtained from a peripheral vein via a preplaced long catheter to measure glyburide concentration and evaluate blood glucose concentrations at 1, 2, 3, 4, 5, 6, 8, 10, 12, 14, 16, 20, and 24 h after drug administration. Before each blood sample was obtained an aliquot of 5 mL blood was withdrawn through the catheter into a heparinized syringe, retained, and then injected back into the dog after the test sample had been obtained. Each 1mL blood sample was tested for blood glucose concentration using a glucose test strip (AlphaTrak2). In this study (because of different IACUC recommendations at different institutions, which also altered our dog weight inclusion criteria), if a value of less than 60 mg/dL was recorded, then intravenous glucose was administered as required to normalize the glucose to within the laboratory reference interval (76–119 mg/dL). The remainder of the sample was placed into a heparinized plastic blood collection tube. This second aliquot was centrifuged at 2,500 G for 15 min and then the plasma was decanted into microcentrifuge tubes and frozen at $-80\,°C$ until analyzed.

When all plasma samples had been collected from all six dogs, glyburide concentration was measured by liquid chromatography/tandem mass spectrometry (LC/MC/MC). Using 300 µL negative canine plasma, the calibration curve was spiked accordingly, with calibrator concentrations of 1, 5, 10, 20, 50, 100, 250, and 500 ng/mL. Sample aliquots of 300 µL were used, and 50 µL glipizide were added to all samples as the internal standard. Each sample was diluted with 1,700 µL PBS Buffer, pH 7.0, making a final volume of 2 mL. Samples were vortexed and allowed to rest at room temperature for approximately 5 min before solid phase extraction (SPE).

Samples were extracted by SPE using the SPEWare CEREX48 Processor (SPEWare Corp., Baldwin Park, CA, USA). Water Wettable Polymer (WWP) SPE cartridges (SPEWare # 12-170418) were used, conditioned with 1 mL methanol, 1 mL RO-DI water, and 1 mL PBS Buffer, pH 7.0. Samples were added to cartridges and allowed to filter through at 1–2 mL/min. Cartridges were washed with 1 mL RO-DI water and dried at full pressure (80 psi) for approximately 10 min. Samples were eluted with 1 mL methanol and dried to a residue under nitrogen at 40 °C. They were reconstituted in 100 µL of a 95:5 reagent grade water:acetonitrile solution before analysis by LC/MS/MS.

Samples were analyzed using a Thermo TSQ Endura LC/MS/MS (Thermo Instruments, San Jose, CA, USA) system. The analytes were separated using an Agilent Eclipse Plus C18 $2.1 \times 50$ mm, 1.8 µm column (#959757-902; Agilent Technologies, Santa Clara, CA, USA). The mobile phases were composed of 0.1% formic acid in water (Mobile Phase A) and 0.1% Formic acid in acetonitrile (Mobile Phase B). The gradient began at 25%B and increased to 95%B to 3.0 min with a flow of 250 µL/min, then increased flow to 500 µL/min for 0.7 min. The LC/MS/MS used a HESI ion source in positive ion mode and a mass resolution of 0.7 on both quadrupoles. Table S1 summarizes the mass spectrometer conditions.

### Pharmacokinetic calculations
Non-compartment and compartmental analyses were performed to provide estimates of various pharmacokinetic parameters. Non-compartmental analysis makes no assumption

about the underlying model of disposition behavior, so it is useful as a descriptive approach. However, compartmental models can be more useful for making predictions and for explaining observed drug disposition characteristics. Therefore, in this study, we performed both to provide both approaches to estimating pharmacokinetic parameters.

Non-compartmental analysis was performed to estimate various pharmacokinetic parameters of glyburide for each individual animal. The following parameters were estimated: time of observed peak plasma drug concentration ($T_{max}$), observed peak drug concentration ($C_{max}$), apparent elimination half-life ($t_{1/2}$, calculated as $\ln(2)/\lambda_z$, $\lambda_z$ being the first-order rate constant associated with the terminal portion of the time-concentration curve as estimated by linear regression of time versus log concentration) area under the plasma concentration-time curve calculated to the last measured concentration ($AUC_{0\text{-last}}$, calculated by the linear trapezoidal rule), and that from time zero extrapolated to infinity ($AUC_{0\text{-inf}}$, calculated by adding the last observed concentration divided by $\lambda_z$ to the $AUC_{0\text{-last}}$), area under the moment curve from time zero to last observed concentration ($AUMC_{0\text{-last}}$), area under the moment curve from time zero extrapolated to infinity ($AUMC_{0\text{-inf}}$), mean resident time estimated using time zero to last observed concentrations ($MRT_{0\text{-last}}$, calculated as $AUMC_{0\text{-last}}/AUC_{0\text{-last}}$), and mean residence time estimated using time zero to infinity ($MRT_{0\text{-inf}}$, calculated as $AUMC_{0\text{-inf}}/AUC_{0\text{-inf}}$).

Compartmental modeling was also attempted to estimate the pharmacokinetic parameters in plasma for each individual animal. One- and two-compartment models were attempted, and Akaike's Information Criterion and visual assessment of observed versus predicted values were used to select the best fit. The following parameters were estimated for each animal: observed time of peak plasma drug concentration ($T_{max}$), observed peak drug concentration ($C_{max}$), apparent absorption and elimination rate ($K$), apparent absorption and elimination half-life ($t_{1/2}$), area under the time-concentration curve (AUC), time from drug administration to onset of absorption ($T_{lag}$), volume of distribution corrected for bioavailability ($V/F$), and clearance corrected for bioavailability ($Cl/F$). All analyses were performed in industry-standard pharmacokinetic software (WinNonLin 7.0.0.2535; Pharsight, Princeton, NJ, USA).

Finally, the available data were used to estimate the drug dosage regimen needed to achieve blood concentrations of glyburide between 25 and 50 ng/mL for at least 72 h. We stipulated that, for convenience of subsequent clinical use of this drug, dosing could not be more frequent than every 8 h. Nonparametric superpositioning in Phoenix WinNonLin was used to create a graph of the mean estimated concentrations of glyburide for 72 h of a dosing regimen consisting of a loading dose of 150 mcg/kg, followed by 75 mcg/kg every 8 h in Dogs 2–5 (the observed data from these dogs had reasonable terminal slopes from which to calculate the elimination rate).

## RESULTS

### Safety study

Five dogs were included: 5 year-old Bichon Frise mixed breed (10.2 kg), 9 year-old Dachshund (8.1 kg), 11 year-old Cardigan Corgi (14 kg), 6 year-old Dachshund (6.8 kg),

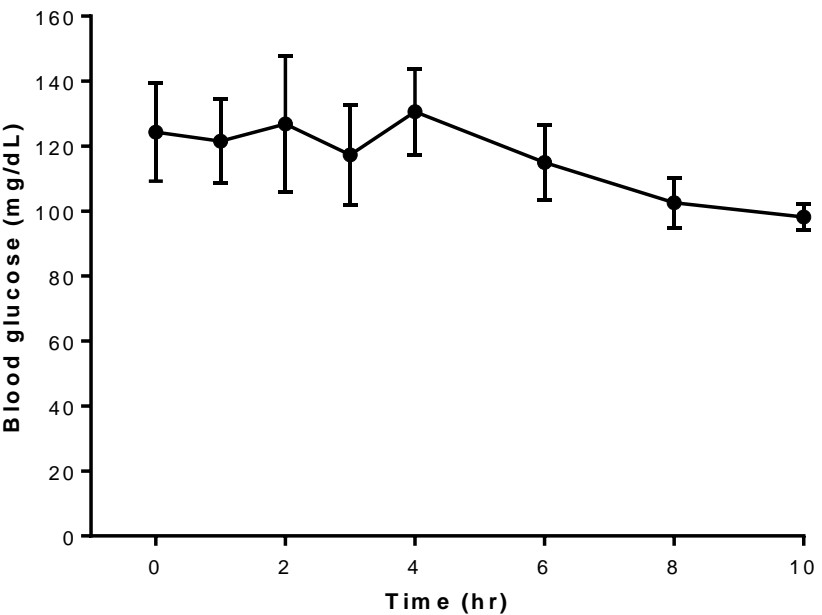

**Figure 1** Relationship between blood glucose concentration and time after oral administration of 75 mcg/kg glyburide at time 0; bars indicate standard error of the mean (s.e.m.).

6 year-old Dachshund (4.2 kg). An initially-enrolled Dobermann was withdrawn from the study at 3 h because a diagnosis of spinal neoplasia was made.

No adverse effects were noted in any dog receiving glyburide. Blood glucose concentrations of less than 50 mg/dL were not observed in any dog, although this exact value was reached in one dog 2 h after receiving the 75 mcg/kg oral dose of glyburide.

Overall, blood glucose concentrations appeared largely unaffected by administration of this dose of glyburide to dogs in this study and remained more-or-less constant throughout the 10-hour period of observation (Fig. 1). Blood concentrations for each dog are shown in Table S2.

## Pharmacokinetic study

Six dogs were included in this part of the study: 6 year-old Dachshund (6.7 kg), 4 year-old German Short-haired Pointer (20.3 kg), 4 year-old French Bulldog (9.2 kg), 10 year-old Shih Tzu (7.8 kg), 5 year-old Dachshund (7.8 kg), 4 year-old Dachshund (7.6 kg). Mean peak glyburide concentration was reached at 13 h after oral administration, and peak concentrations exceeded 25 ng/mL in four of the six dogs but did not reach this concentration in the remaining two (Dog 1 and Dog 6). Values exceeding 25 ng/mL were detected once each in Dog 2 and Dog 4, twice in Dog 5 and five times in Dog 3 (Table S3).

Serum concentration data and pharmacokinetic parameter estimates suggest that Dog 6 is an outlier: $T_{max}$ was at the last observed time point for this dog (see Table 1), and the plasma concentrations for this dog were much lower than the other dogs at all time points, resulting in a skewing of mean observed serum concentrations (Fig. 2). In addition, compartmental pharmacokinetic analysis was not possible for Dog 6 with any of the

**Table 1** Estimates of pharmacokinetic parameters of glyburide in dogs ($n = 6$) based on non-compartmental and compartmental analysis of serum concentration after one oral dose of glyburide (75 mcg/kg) (see text for abbreviations; SD, standard deviation).

| Parameter | Units | Dog 1 | Dog 2 | Dog 3 | Dog 4 | Dog 5 | Dog 6 | Mean | SD |
|---|---|---|---|---|---|---|---|---|---|
| **Non-compartmental analysis** | | | | | | | | | |
| $T_{max}$ | hr | 10 | 10 | 12 | 2 | 12 | 24 | 12 | 7 |
| $C_{max}$ | ng/mL | 20.9 | 26.2 | 54.3 | 32.2 | 44.2 | 9.7 | 31.3 | 14.7 |
| $\lambda_z$ | /hr | 0.075 | 0.063 | 0.142 | 0.110 | 0.127 | [a] | 0.103 | 0.030 |
| $t_{1/2\lambda z}$ | hr | 10.9 | 10.9 | 4.9 | 6.3 | 5.4 | [a] | 7.7 | 2.7 |
| $AUC_{0-last}$ | hr ng/mL | 272.5 | 343.9 | 526.4 | 212.5 | 390.7 | 115.05 | 310.1 | 131.3 |
| $AUC_{0-inf}$ | hr ng/mL | 352.2 | 526.9 | 576.5 | 229.8 | 437.8 | [a] | 424.6 | 124.0 |
| AUC % Extrap | % | 22.6 | 34.7 | 8.7 | 7.5 | 10.8 | [a] | 16.9 | 10.4 |
| $AUMC_{0-last}$ | hr ng ng/mL | 3,611 | 4,622 | 5,750 | 1,499 | 4,968 | 1,887 | 3,723 | 1,570 |
| $AUMC_{0-inf}$ | hr ng ng/mL | 6,583 | 11,905 | 7,308 | 2,071 | 6,469 | [a] | 6,867 | 3,124 |
| $MRT_{last}$ | hr | 13.3 | 13.4 | 10.9 | 7.1 | 12.7 | 16.4 | 12.3 | 2.8 |
| $MRT_{0-inf}$ | hr | 18.7 | 22.6 | 12.7 | 9.0 | 14.8 | [a] | 15.6 | 4.7 |
| **One-compartment model**[b] | | | | | | | | | |
| $T_{max}$ | hr | 12.1 | 20.7 | 7.1 | 1.7 | 17.5 | [a] | 12 | 7 |
| $T_{lag}$ | hr | – | – | 0.8 | – | – | [a] | | |
| $C_{max}$ | ng/mL | 9.7 | 14.9 | 28.8 | 19.7 | 8.6 | [a] | 16.4 | 7.4 |
| $K$ | 1/hr | 0.0829 | 0.0484 | 0.1566 | – | 0.0571 | [a] | 0.0862 | 0.0426 |
| $K01$ | 1/hr | – | – | – | 1.6933 | – | [a] | | |
| $K10$ | 1/hr | – | – | – | 0.1203 | – | [a] | | |
| $t_{1/2}K10$ | hr | – | – | – | 0.4 | – | [a] | | |
| $t_{1/2}K01$ | hr | – | – | – | 5.8 | – | [a] | | |
| $t_{1/2}$ | hr | 8.4 | 14.3 | 4.4 | – | 12.1 | [a] | 9.8 | 3.8 |
| AUC | hr ng/mL | 317.7 | 837.7 | 500.6 | 200.5 | 411.0 | [a] | 453.5 | 216.4 |
| $V/F$ | mL/kg | 2,848.9 | 1,850.7 | 956.4 | – | 3,198.0 | [a] | 2,213.5 | 878.2 |
| $CL/F$ | mL/hr/kg | 236.1 | 89.5 | 149.8 | 374.0 | 182.5 | [a] | 206.4 | 96.3 |

**Notes.**

[a] See text for explanation about incomplete pharmacokinetic analysis of these dogs.

[b] One-compartment models with observed values weighted by $1/y^2$ and first order input and elimination fit best for Dogs 1–5; the model selected for Dogs 1, 2, and 5 included no lag time and assumption of $K10 = K01$; the model selected for Dog 3 was similar with the exception of a lag time; the model selected for Dog 4 included no assumptions about K10 and no lag time.

attempted models. In the non-compartmental analysis, two dogs (Dog 1 and Dog 2) required higher than desirable extrapolation percentages to calculate AUC: both were greater than 20%, the recommended maximal extrapolation percentage.

### Relationship between glyburide and glucose concentrations in pharmacokinetic study

Blood glucose concentration ranged from 50 to 324 mg/dL (Table S4). In one dog the blood glucose concentration reached 50 mg/dL at 4 h after drug administration, which was lower than the pre-defined intervention level of 60 mg/dL for this study, and intravenous glucose supplementation was given; subsequent recordings of blood glucose concentrations were excluded in the analysis of glucose *versus* glyburide concentration for

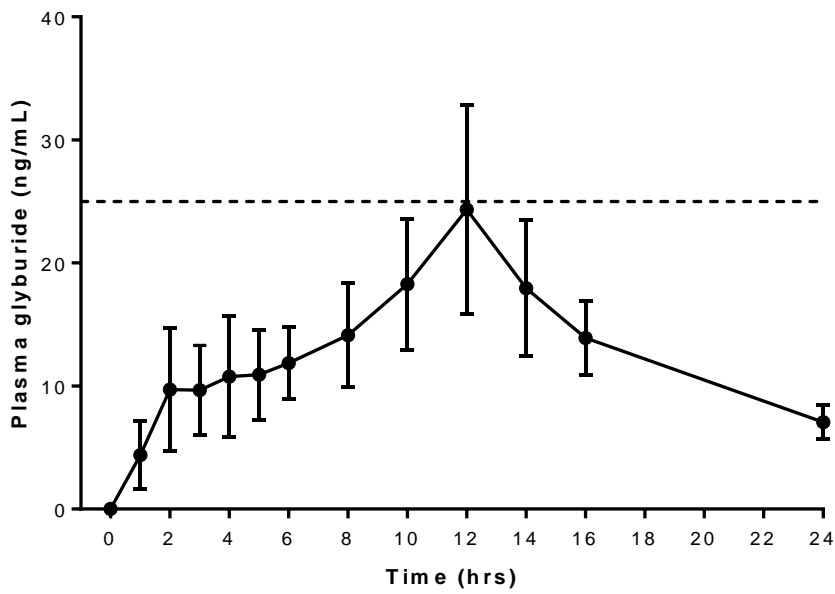

**Figure 2** **Relationship between serum glyburide concentration and time after oral administration of 75 mcg/kg glyburide at time 0; bars indicate s.e.m.** Dashed horizontal line indicates desired minimum therapeutic concentration (see text).

this dog. Linear regression analysis and plot (Fig. 3) of the relationship between drug and glucose concentration in our whole study population suggested there was not a significant association between these variables at this glyburide dosage ($R^2 = 0.008$; $P = 0.464$).

Superpositioning of observed concentrations in Dogs 2–5 using a loading dose of 150 mcg/kg glyburide followed by 8-hourly dosing at 75 mcg/kg produced an average plasma level of above 25 ng/mL, but below 50 ng/mL, for most of a simulated 72-hour treatment period (Fig. 4).

## DISCUSSION

We show here that our conservatively-selected glyburide dosage of 75 mcg/kg achieved the target blood level of between 25 and 50 ng/mL in only four of six dogs in our pharmacokinetic study, and the periods for which this was achieved were brief. On the other hand, this dose did not cause unsafe reductions in blood glucose concentrations in any of the dogs. In two dogs there was a reduction in blood glucose to <60 mg/dL but neither of these animals exhibited any ill-effects, although they were anesthetized at the time. It seems likely that these periods of hypoglycemia were not the result of glyburide itself since there appeared to be no relationship between glyburide and glucose blood concentrations in these dogs at this dosage, suggesting that glyburide can be safely administered at this, or higher, doses, as previously reported (http://products.sanofi.ca/en/diabeta.pdf). All 11 dogs recruited to this study uneventfully recovered the ability to walk after surgery.

In a clinical trial evaluating glyburide for acute spinal cord injury, because it is a rapidly evolving condition, there is a need to achieve effective blood concentrations as rapidly

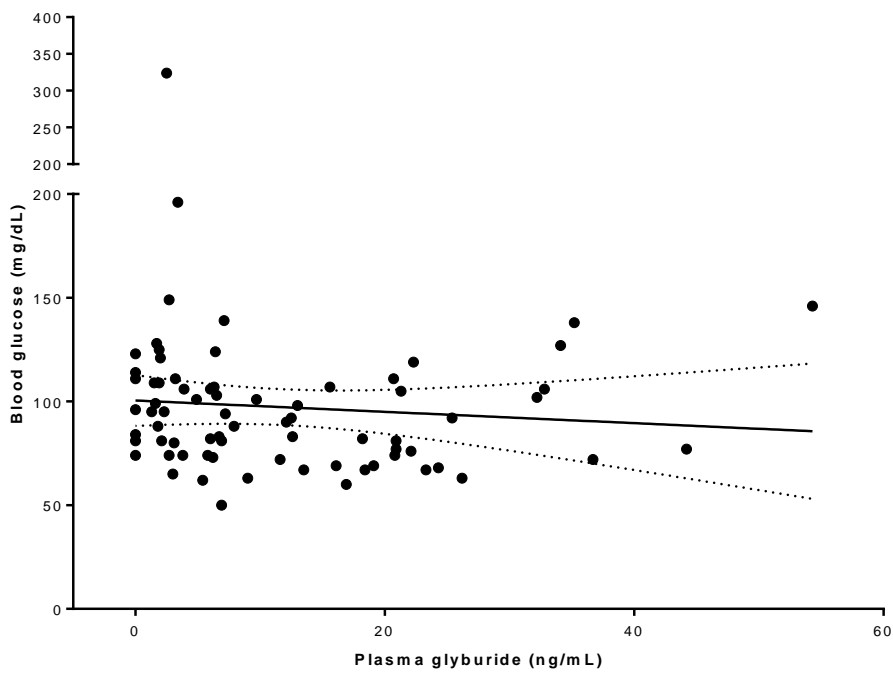

**Figure 3** **Relationship between blood glucose and serum glyburide concentrations.** The line indicates linear regression line of best fit (dotted lines indicate 95% confidence intervals) implying that these variables are not significantly associated (for glyburide concentrations in this range); $R^2 = 0.008$; $P = 0.464$.

as possible and maintain these for at least 72 h (in line with other clinical uses of this drug; *Sheth et al., 2016a*; *Sheth et al., 2016b*). We estimate that a safe and effective dose regimen would be a loading dose of 150 mcg/kg, followed by 75 ng/kg every 8 h. By extrapolation from the data we have collected in this study such a regimen is expected to lead to therapeutic concentrations (>25 ng/mL; see *Simard et al., 2008*) 2–3 h after dosing and for up to for 72 h. The loading dose is proposed as a means to ensure that the drug attains a therapeutic concentration as soon as possible after oral dosing. The assumption in treatment of spinal cord injury is that rapid treatment is essential for effective neuroprotection. Although intravenous preparations (see *Sheth et al., 2014*) may be preferable for this reason, in veterinary medicine much of the delay between onset of injury and initiation of treatment is caused by delay in owner recognition of the condition, travel to a specialist clinic, triage and obtaining owner consent for treatment, which cannot easily be eliminated. Fortunately, although the stimulus for the secondary injury mechanisms of spinal cord injury is the moment of impact, it can take hours to days for the tissue-destructive mechanisms to become fully up-regulated (*Crowe et al., 1997*). Indeed, Sur1, the target of glyburide, becomes increasingly widely expressed during the first 24 h after experimental spinal cord injury (*Simard et al., 2007*), suggesting that, although earlier administration is likely to be superior, attaining appropriate plasma drug concentrations within 2–3 h after examination at a veterinary clinic may still be efficacious.

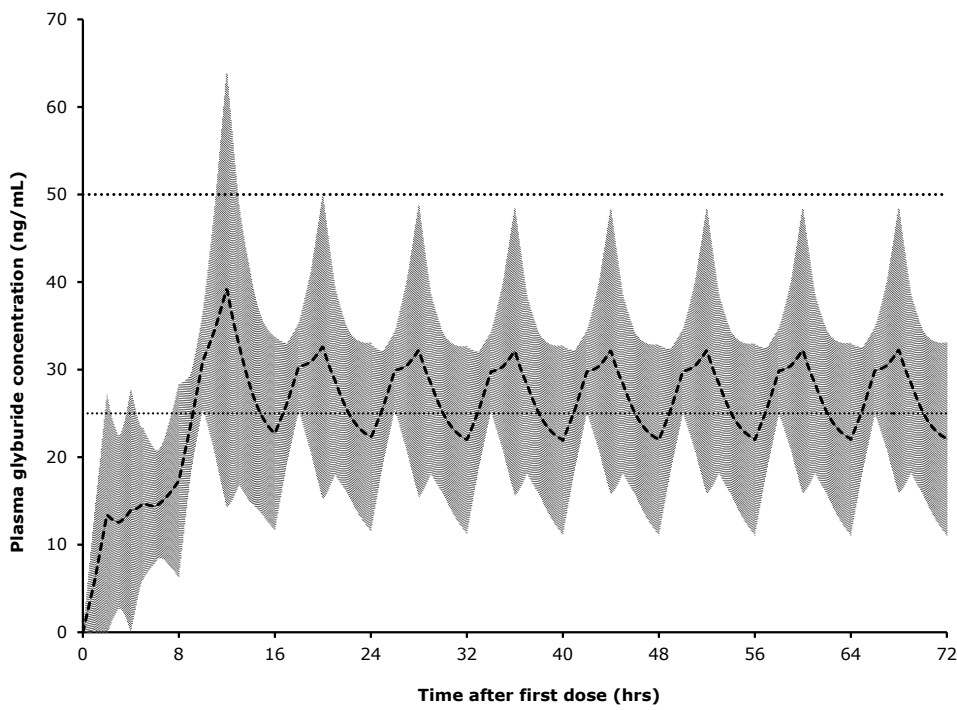

**Figure 4   Predicted glyburide concentrations based on superpositioning of observed concentrations in individual dogs.** Shaded area represents one standard deviation from mean.

An important aspect of this current study is that we used a sample of dogs similar to those that will be targeted by this therapy in a clinical trial. Although such cases will produce more heterogeneous pharmacokinetic data than normal, conscious, laboratory animals, it is important to assess drug disposition and adverse effects on blood glucose that could possibly occur in this category of veterinary patient. Specifically, dogs undergoing anesthesia for imaging and surgery are routinely administered intravenous fluids that by expanding the circulating volume may affect drug concentration and, in addition, may receive other drugs including antibiotics, analgesics and gaseous anesthetic agents, all of which may alter glyburide concentration in blood or its effects on blood glucose. Our study shows that despite these factors the dose of 75 mcg/kg can achieve sufficiently high concentrations to affect function of the Sur1 channel (see *Simard et al., 2008*) and has no detectable adverse effects. Our data also imply that the dose will need to be scaled-up to achieve rapid and maintained plasma concentrations adequate to achieve the desired clinical benefit.

## CONCLUSIONS

This study shows that glyburide given orally at 75 mcg/kg only just achieves blood concentrations of the drug appropriate for targeting function of the Sur1 channel in cells in the injured spinal cord but does not appear to adversely affect blood glucose concentrations. To achieve rapid and appropriate concentrations of glyburide we suggest

using an initial loading dose of 150 mcg/kg followed by repeat dosing at 75 mcg/kg every 8 h.

## ACKNOWLEDGEMENTS

We thank the 'Dogs Helping Dogs' laboratory at Texas A&M University for access to their equipment for blood sample processing.

### Funding

The authors received no funding for this work.

### Competing Interests

The authors declare there are no competing interests.

### Author Contributions

- Nick Jeffery, Megan Konarik, Travis Mays and Virginia Fajt conceived and designed the experiments, performed the experiments, analyzed the data, contributed reagents/materials/analysis tools, prepared figures and/or tables, authored or reviewed drafts of the paper.
- C. Elizabeth Boudreau conceived and designed the experiments, authored or reviewed drafts of the paper.

### Animal Ethics

The following information was supplied relating to ethical approvals (i.e., approving body and any reference numbers):

Dogs included in this study presented for treatment of acute spinal cord injury to the small animal hospital of the Colleges of Veterinary Medicine at Iowa State University or Texas A&M University, and study protocols were approved by the relevant Institutional Animal Care and Use Committees (Iowa State, log number: 1-16-8148-K; Texas A&M, IACUC 2016-0324 CA, reference number 044949).

### Data Availability

All raw data is supplied as Tables in the Supplementary Information.

### Supplemental Information

Supplemental information for this article can be found online at http://dx.doi.org/10.7717/peerj.4387#supplemental-information.

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
