# Peer review of "Pharmacokinetics and safety of oral glyburide in dogs with acute spinal cord injury"

_PeerJ, doi:10.7717/peerj.4387_

## Round 0.1 · original submission · Major Revisions

Overall, we think it is an interesting study but some concerns need to be solved before consideration for acceptance. The dose issue pointed out by the Reviewer 1 need to be addressed. Both reviewer 1 and 3 have some concerns on your experimental design, which need further clarification.

Reviewer 1 ·

Basic reporting

This manuscript is an experiment report rather than a scientific paper.
There are several concerns when I was reviewing this manuscript:

1. The innovation and motivation of this study are not clear.
2. The background of this study is not well introduced.
3. The reason to choose dog but not other species to conduct this study is not clear.
4. How to define the dose applied in this study is not clear.

Experimental design

1. In the "Method", the dose used in this study is 75 mcg/kg, but in the "Conclusion" (line 323-327), the dose is 75 ng/kg.
2. In line 176, the quantification range should be from 10 - 500 ng/mL, but in Figure 2, the concentration level at first time point is below 10 ng/mL.
3. In table 1, both non-compartment and one-compartment models are used to calculate the PK parameters, but the authors are not telling people that which one is better, and why.
4. For the blood glucose testing, we do not know that the dogs are fasting or being given food during the study.
5. If the dogs are given food, the study design for this manuscript is not sufficient, since both food and drug may have effect on glucose level.

Validity of the findings

Conclusion is not well stated.

Additional comments

The authors should be enhanced and be more professional in the PK area.

·

Basic reporting

All criteria of basic reporting are fully met.

Experimental design

All criteria of experimental design are fully met.

Validity of the findings

The data re robust and the conclusions are appropriate

Additional comments

This is a study of glyburide safety and pharmacokinetics in dogs with spinal cord injury. The manuscript is very well written with appropriate citations, the rational for the study is clear, the methods are appropriately detailed, the results are clearly presented, and the discussion lists appropriate caveats and shortcomings. I unequivocally endorse publication.

I encountered only a single difficulty that can be addressed with minor revision: in two instances, there is reference to "negative" canine plasma. It is unclear to me what the term negative implies here.

·

Basic reporting

The article was written clearly in professional English, well organized and easy to follow. The authors are describing clear background of their purpose of study with proper references. However, some information and raw data might be needed.

Line 95; ‘because these other interventions and the lesion itself may alter blood drug concentrations or risk of adverse effect’.
Please mentioned whether the authors did investigate severity of injury would alter because dogs with different severity are included in Study 2 (e.g. paraparesis and paraplegia).

Line 234~&249~; Why the authors described the details of patients including age, dog bleed and body weight in Study 1 which is not included in Study 2? It is not clearly explained the omission in Study 2. Please mention why did the authors put those information in study 1 and did not in study 2.

Table 1 and supplemental table, the authors provided concentration of blood glucose level of dogs in Study 1 but not supplied those data in Study 2. Therefore, I cannot see the relation between glyburide concentration and blood glucose level, though the authors described no relation in Study 2.

Experimental design

The study design is clear; the authors investigated i) safety of oral glyburide, ii) pharmacokinetics of the drug after administration, and iii) estimate pharmacokinetic distribution in dogs with spontaneously occurred acute SCI.

However the criteria of selection of dogs and dropping out are not clearly described.
2-i) Animals; Size of dogs
Line 119; Dogs of any size were accepted for inclusion,,,,
The supplemental file (consent form final) says dogs weighing at least 5kg in ‘Eligiblility for participation’,,,, for Study2. I can see Study 1 had dog with less than 5kg and consent form for safety is not including body weight restriction. Therefore the authors may want to explane why they changed the criterion.

2-ii) Animals;
Line 123; Dogs previously diagnosed with hyperadrenocorticism or diabetes mellitus were excluded.
I think these criteria would be reasonable because those diseases might affect to the pharmacokinetics of glyburide and blood glucose level. But I can see dog 5 in study 1 had relatively high blood glucose level, and in the result section (line 263) the authors described as ‘Blood glucose concentration ranged from 50 to 324 mg/dL,,,’ in study 2. The result of 324mg/dL seems to be abnormally high concentration of blood glucose.
My questions are
- Why Dog 5 in study 1 had relatively high blood glucose?
- 324mg/dL of blood glucose would be quite high,,,, so it would be better to explain the reason.

2-iii) Methods; Level of blood glucose
The authors set different minimum levels of blood glucose in criteria of Study 1 (50mg/dL) and Study 2 (60mg/dL). Judging from the provided reference interval of glucose level in dogs (76-119mg/dL), 50mg/dL to be low level.
The questions are;
- Why the authors set different minimum levels of blood glucose in study 1 and 2?
- May better to describe the concentration around 50mg/dL of glucose level would be safe in canine patients.

2-iv) Methods; 2. Pharmacokinetic study
Line 162; Bllod samples (1mL) were obtained from,,,,, after drug administration. Before each blood sample was obtained an aliquot of 5ml blood was withdrawn through the catheter and discarded.
In this study, the authors evaluated the blood samples at 13 time points during 24 hours (is this correct?). At each time point 6ml of blood was taken from dog, and in total 78ml of blood was taken from one dog during 24 hours? Is this volume of blood loss safe for small dogs (5kg of body weight)?

Validity of the findings

The data in this manuscript are not assessed previously and have valuable impacts on future studies in this field. Discussion and conclusion are focused on their primary question and supporting their results.

Additional comments

Abstract
Line 34; paraparetic
I can see dogs in study 1 showed paraplegia and dogs in study 2 showed either paraparesis or paraplegia. I am not sure whether ‘paraparetic’ would be a best word to represent these conditions.

Materials & Methods
Line 104; they receive intravenous fluid infusions
Did all dogs have same fluid infusions at same rate which might be affecting to both glucose level and concentration of glyburide?

Discussion
Line 274
I can see some dogs had exceeded level (more than 50ng/mL) of glyburide concentration at peak (Cmax). Does this concentration have any side effect to dogs? If there is an established safety range of the drug it might be better to mention.

Line277
Though I am not sure what level of blood glucose would be unsafe in dogs, around 50mg/dl seems to be unsafe for me even if dogs did not show any clinical symptom. Therefore, the authors might be better to describe why the concentration is estimated as safe in dogs.

---

## Round 0.2 · accepted · Accept

After contingent assessments by the expert reviewers and editors we think the manuscript has been greatly improved and the reviewers' concerns are largely resolved. We are delighted to inform you that the manuscript "Pharmacokinetics and safety of oral glyburide in dogs with acute spinal cord injury " has been accepted for publication in PeerJ. Thank you for allowing us to review this interesting work for PeerJ.

·

Basic reporting

The manuscript was written clearly in professional English, well organized and easy to follow. The authors are describing clear background of their purpose of study with proper references. All clear now with newly added supplemental files. Thank you for making the files.

Experimental design

The study design is clear; the authors investigated i) safety of oral glyburide, ii) pharmacokinetics of the drug after administration, and iii) estimate pharmacokinetic distribution in dogs with spontaneously occurred acute SCI. In addition the authors have added details of methods in this revised version.

Validity of the findings

The data in this manuscript are not assessed previously and have valuable impacts on future studies in this field. Discussion and conclusion are focused on their primary question and supporting their results.

Additional comments

Dear Authors
Thank you for making great revison on the manuscript according to reviewers comments. I think the manuscript has been improved with clear objectcs, methods, results and discussion. I am pleased seeing revised version of the manuscript and do not have further concern. Thanks.
Best regards
.